# Lung Ultrasound to Determine the Effect of Lower vs. Higher PEEP on Lung Aeration in Patients without ARDS—A Substudy of a Randomized Clinical Trial

**DOI:** 10.3390/diagnostics13121989

**Published:** 2023-06-07

**Authors:** Claudio Zimatore, Anna Geke Algera, Michela Botta, Charalampos Pierrakos, Ary Serpa Neto, Salvatore Grasso, Marcus J. Schultz, Luigi Pisani, Frederique Paulus

**Affiliations:** 1Department of Intensive Care, Amsterdam University Medical Centers, Location AMC, 1105 AZ Amsterdam, The Netherlands; a.g.algera@amsterdamumc.nl (A.G.A.); m.botta@amsterdamumc.nl (M.B.); charalampos_p@hotmail.com (C.P.); ary.serpaneto@monash.edu (A.S.N.); marcus.jschultz@gmail.com (M.J.S.); luigipisani@gmail.com (L.P.); f.paulus@amsterdamumc.nl (F.P.); 2Department of Emergency and Organ Transplantation, University of Bari Aldo Moro, 70124 Bari, Italy; salvatore.grasso@uniba.it; 3Laboratory of Experimental Intensive Care and Anesthesiology (LEICA), Amsterdam University Medical Centers, Location AMC, 1105 AZ Amsterdam, The Netherlands; 4Department of Intensive Care, Brugmann University Hospital, Université Libre de Bruxelles, 1050 Brussels, Belgium; 5Department of Critical Care Medicine, Hospital Israelita Albert Einstein, São Paulo 05652-900, Brazil; 6Australian and New Zealand Intensive Care Research Centre (ANZIC-RC), School of Public Health and Preventive Medicine, Monash University, Melbourne 3000, Australia; 7Mahidol–Oxford Tropical Medicine Research Unit (MORU), Mahidol University, Bangkok 10400, Thailand; 8Nuffield Department of Medicine, Oxford University, Oxford OX3 7FZ, UK; 9Department of Anesthesia and Intensive Care, Miulli General Hospital, 70021 Acquaviva delle Fonti, Italy; 10ACHIEVE, Centre of Applied Research, Faculty of Health, Amsterdam University of Applied Sciences, 1091 GC Amsterdam, The Netherlands

**Keywords:** intensive care, critical care, ventilation, positive end–expiratory pressure, PEEP, invasive ventilation, lung aeration, ultrasound, LUS, LUS score

## Abstract

Background: Ventilation with lower positive end–expiratory pressure (PEEP) may cause loss of lung aeration in critically ill invasively ventilated patients. This study investigated whether a systematic lung ultrasound (LUS) scoring system can detect such changes in lung aeration in a study comparing lower versus higher PEEP in invasively ventilated patients without acute respiratory distress syndrome (ARDS). Methods: Single center substudy of a national, multicenter, randomized clinical trial comparing lower versus higher PEEP ventilation strategy. Fifty–seven patients underwent a systematic 12–region LUS examination within 12 h and between 24 to 48 h after start of invasive ventilation, according to randomization. The primary endpoint was a change in the global LUS aeration score, where a higher value indicates a greater impairment in lung aeration. Results: Thirty–three and twenty–four patients received ventilation with lower PEEP (median PEEP 1 (0–5) cm H_2_O) or higher PEEP (median PEEP 8 (8–8) cm H_2_O), respectively. Median global LUS aeration scores within 12 h and between 24 and 48 h were 8 (4 to 14) and 9 (4 to 12) (difference 1 (–2 to 3)) in the lower PEEP group, and 7 (2–11) and 6 (1–12) (difference 0 (–2 to 3)) in the higher PEEP group. Neither differences in changes over time nor differences in absolute scores reached statistical significance. Conclusions: In this substudy of a randomized clinical trial comparing lower PEEP versus higher PEEP in patients without ARDS, LUS was unable to detect changes in lung aeration.

## 1. Introduction

Critically ill patients frequently need intubation for invasive ventilation or airway protection. Despite the absence of evidence for the benefits of this approach, higher PEEP is increasingly used, not merely in patients with acute respiratory distress syndrome (ARDS), but also in patients without this complication [1,2,3]. A recent randomized clinical trial, named ‘REstricted vs. Liberal Positive End–expiratory Pressure in patients without acute respiratory distress syndrome (ARDS)’ (RELAx), showed a lower PEEP (0 to 5 cm H_2_O) ventilation strategy to be non–inferior to ventilation with higher PEEP (8 cm H_2_O) with respect to the number of ventilator–free days in patients without ARDS [4]. This study also showed that ventilation with lower PEEP was associated with worse oxygenation and a higher occurrence of desaturations. This at least suggest that atelectasis developed more often in patients who were ventilated with lower PEEP.

Changes in lung aeration might be detectable with lung ultrasound (LUS) [5]. Indeed, LUS scores have been used successfully to detect changes in regional or global lung aeration in previous studies [6,7]. Recently, a rise in the LUS score, suggesting greater impairment in aeration, was found to have associations with unsuccessful liberation from ventilation [8].

The aim of this study was to determine if LUS is able to detect differences in aeration between patients ventilated with a higher and a lower PEEP strategy. In a substudy of the abovementioned RELAx, we performed LUS at different time points, and hypothesized that the lower PEEP strategy would result in higher global LUS scores compared to the higher PEEP strategy. We also tested the capability of a regional LUS score to detect local differences in aeration, and a score that focuses on the presence of atelectasis.

## 2. Materials and Methods

### 2.1. Study Design and Patients

RELAx was a national, multicenter, randomized clinical trial comparing a higher PEEP with a lower PEEP strategy in invasively ventilated ICU patients without ARDS. The study protocol of this study was approved by the Institutional Review Boards of all participating hospitals. The protocol was prepublished [9], and the study was registered at clinicaltrials.gov (study identifier NCT03167580). Patients were randomized for the seminal trial in a 1:1 ratio to a lower or higher PEEP strategy group. The local investigators performed randomization using a central, dedicated, password-protected, encrypted, web-based automated randomization system (SSL-encrypted website with ALEA software, TenALEA Consortium). RELAx used a deferred informed consent procedure. The results of this study have been reported elsewhere [4].

In this single center substudy, we evaluated the capability of LUS to detect differences in aeration between patients ventilated with a higher versus a lower PEEP strategy. The protocol of this substudy was approved by the Institutional Review Board of the Amsterdam University Medical Centers, location ‘AMC’, Amsterdam, The Netherlands. The substudy was registered with the parent study at clinicaltrials.gov under the same identifier. For this substudy we needed written informed consent.

The parent study and this substudy were supported by a grant from ZonMW (the Netherlands Organization for Health Research and Development) and the Amsterdam University Medical Centers, Location AMC. The funders had neither a role in the design and conduct of RELAx, nor in the interpretation of the data and the preparation of the final report.

Patients were eligible for participation in RELAx if they: (1) were aged 18 years or older; (2) were admitted to one of the participating hospitals; (3) needed invasive ventilation for reasons other than ARDS; and (4) expected not to be extubated within 24 h. Patients were excluded if randomization was not possible within the first hour of invasive ventilation in the ICU. Additional major exclusion criteria were pregnancy; a history of chronic obstructive pulmonary disease class III or IV or restrictive pulmonary disease; increased and uncontrollable intracranial pressure or delayed cerebral ischemia; and ongoing cardiac ischemia [4].

For this substudy, we also excluded patients who had evidence for presence of cardiac failure or fluid overload, based on an objective assessment such as echocardiography in the medical record or on judgment of the treating physician.

### 2.2. Ventilation with a Higher versus Lower PEEP

In the RELAx trial, for patients assigned to the higher PEEP group, PEEP was set at 8 cm H_2_O. In patients assigned to the lower PEEP strategy, the lowest possible PEEP level between 0 to 5 cm H_2_O was targeted. Ventilation started with 5 cm H_2_O, and every 15 min PEEP was down–titrated 1 cm H_2_O to 0 cm H_2_O unless hypoxemia developed, which required an increase in the fraction of inspired oxygen (FiO_2_) higher than 60%.

### 2.3. Lung Ultrasound

LUS examination was performed by experienced and trained physicians using a 2–5 MHz convex probe. LUS examination was performed at predefined timepoints: (i) within 12 h after group assignment, and (ii) after 24–48 h.

Each hemithorax was divided into six areas: the anterior, lateral, and posterior areas, each divided into upper and lower quadrants, using the parasternal–, anterior axillary–, posterior axillary–, and the paravertebral–lines as borders. The 12 regions were extensively examined; the worst ultrasound abnormality detected was considered as characterizing the region examined.

Four ultrasound aeration patterns were defined: (A) normal aeration: presence of lung sliding with A lines or ≤2 isolated B lines (0 point); (B1) moderate loss of aeration: artefacts occupying ≤50% of the pleura (1 point); (B2) severe loss of lung aeration: artefacts occupying >50% of the pleura (2 points); (C) consolidated lung tissue: hypoechoic or tissue–like area (3 points) (Appendix A) [10]. Individual global LUS score (0–36) was calculated as the sum of the 12 quadrants score, with higher scores indicating more severe aeration loss. In case a region could not be scanned, or the quality of the image was insufficient for scoring, the missing value for that region was replaced with the mean value of the other zones from the same examination [11].

Regional LUS scores (0–6) were calculated as the sum of anterior, lateral, and posterior quadrants. The ultrasound reaeration score is an alternative calculation that was assessed from changes in the ultrasound pattern of each region examined [5,12]. An increase in the reaeration score indicates an increase in aeration. The presence of additional sonographic signs previously described for atelectasis were reported for each of the 12 regions examined, including subpleural consolidations and presence of air bronchograms in consolidated areas. In particular, static air bronchograms were assessed, as these are more frequently observed in atelectasis while dynamic air bronchograms characterize pneumonia-related consolidations [13]. Numerical scores were calculated for subpleural consolidations (SPC), static air bronchograms and B–lines > 2; i.e., each lung ultrasound region with the presence of SPC, static air bronchograms or B–pattern was assigned a 1 and all 12 zones were summed, resulting in a score from 0 to 12 per patient, with higher scores suggesting presence of more atelectasis.

### 2.4. Blinding and Masking

The attending nurses and physicians could not be kept blind for the intervention. The investigator who performed LUS, however, remained unaware of group assignment.

### 2.5. Endpoints

The primary endpoint was the change in global LUS score in the first 48 h of ventilation, i.e., from the first to the second LUS examination. Secondary endpoints were changes in regional LUS scores, the reaeration score, and presence of atelectasis.

### 2.6. Power Calculation

Considering a baseline LUS score between 5 and 10 in ICU patients without ARDS, we expected a clinically significant difference in the change in global LUS score of 2.5 points between the lower and higher PEEP groups, a standard deviation of the global LUS score of 3 points, an alpha of 0.05, and a power of 0.8. We calculated that a minimum of 23 patients per group were needed.

### 2.7. Statistical Analysis

Categorical variables are reported as numbers and percentages and continuous variables are presented as medians with interquartile ranges (IQRs). The comparison of continuous variables between the two groups was performed using the independent samples *t*-test in case of a normal distribution, otherwise the Mann–Whitney U test was used.

Comparisons are shown with the Hodges–Lehmann estimate of the median difference and 95% CI. The comparisons of categorical variables between both groups were performed using the chi–square test.

A two–sided *p* value < 0.05 was considered statistically significant with exact *p* values given unless *p* < 0.001. All analyses were performed with use of R software, version 3.6.3 (R Core Team, 2016, Vienna, Austria); graphs were constructed using GraphPad Prism 9.4 for Windows, GraphPad software, www.graphpad.com, accessed on 5 April 2023.

## 3. Results

### 3.1. Patients

Between 24 July 2018 and 15 December 2019, a total of 146 patients were enrolled in RELAx in the Amsterdam UMC, location AMC. Of 94 patients who underwent LUS examination within 12 h after enrollment in RELAx, 57 patients also underwent LUS examination after 24–48 h. Thus, data of 57 patients could be used: 24 patients assigned to ventilation with higher PEEP and 33 patients assigned to ventilation with lower PEEP (Figure 1).

Patient demographics and ventilation characteristics are presented in Table 1 and Table 2. There were no significant differences between the lower PEEP and higher PEEP group regarding demographics, co–morbidities, baseline fluid balances, and reasons for intubation and ventilation. There was no difference between the groups in the frequency of pneumonia or cardiogenic pulmonary edema diagnosis. The most frequent reasons for invasive ventilation were acute hypoxemic respiratory failure and depressed consciousness.

Patients in the lower PEEP group were ventilated with median PEEP of 1 (0–5) cm H_2_O; patients in the higher PEEP group were ventilated with median PEEP of 8 (8–8) cm H_2_O (Table 2). Contrast in PEEP remained over the first 48 h of ventilation. Patients in the lower PEEP group were ventilated with higher FiO_2_ and had a less positive fluid balance at the second LUS examination compared to patients in the higher PEEP group. The PaO_2_/FiO_2_ ratio was higher in the high PEEP group in both LUS timepoints, though this finding was not statistically significant. None of the study patients received prone positioning.

### 3.2. Lung Ultrasound

Median global LUS aeration score within 12 h and between 24 and 48 h was 8 (4 to 14) and 9 (4 to 12) (difference 1 (–2 to 3)) in the lower PEEP group, and 7 (2–11) and 6 (1–12) (difference 0 (–2 to 3)) in the higher PEEP group (Figure 2). Changes in global LUS score were not different between patients in the lower PEEP group and patients in the higher PEEP group.

Differences in changes in regional LUS scores, in the reaeration score, and the presence of atelectasis also did not reach statistical significance. Static air bronchograms were present in 9.1% of patients in the lower PEEP group and 8.3% in the higher PEEP group at baseline (Table 3).

## 4. Discussion

This substudy of a larger randomized clinical trial that tested the noninferiority of a lower PEEP ventilation strategy with a higher PEEP ventilation strategy in invasively ventilated patients without ARDS investigated whether LUS is capable of detecting differences in changes in aeration. The findings suggest that LUS aeration scores are not helpful in detecting differences in change in aeration. The role of a higher positive end-expiratory pressure in patients without ARDS is debated. Several observational studies have demonstrated a change in ventilation strategy, with an increase in PEEP in the last two decades [14,15].

Our study has strengths. In the parent study, patients were ventilated using a pragmatic protocol that was strictly adhered to. To minimize a possible carryover effect, randomization was performed within one hour of start of invasive ventilation. Randomization led to clear and consistent contrast in PEEP between the two randomization groups, and also differences in oxygenation, suggesting that the lower PEEP strategy was associated with development of atelectasis. By performing LUS at two timepoints, we were able to study the changes over time. The analysis of the substudy was preplanned, and we strictly followed the analysis plan that was in place before cleaning and closing of the database. To prevent bias, both the analysis of data and LUS were performed by investigators who remained blinded to the randomization arm, and ventilator parameters were hidden during the LUS exam.

LUS scores were not capable of detecting changes in aeration, even while the difference in PEEP was 7 cm H_2_O between the two groups. Differences in aeration were likely the cause of the observed differences in oxygenation between patients ventilated with lower and higher PEEP. It should be noted, though, that we did not use the gold standard, i.e., chest computed tomography (CT). The reasons why LUS scores could not detect differences in changes in aeration nor even differences in absolute LUS scores between the two groups may include the following: first, differences in changes in aeration may have been negligible between the two groups, and thus rightly not picked up by the LUS score we used; it is also possible, though, that differences in oxygenation came from minor changes in aeration, that are undetectable with LUS. We noted a discrete increase in LUS score in the lower PEEP group, suggesting that LUS did detect a change in lung aeration, but this increase did not reach statistical significance. We cannot exclude the possibility that we were underpowered to detect a change, though.

Whether LUS aeration scores are capable of detecting meaningful changes in ventilated ICU patients is under debate. Studies so far have mostly, if not exclusively, included patients with severe ARDS. In a recent study in patients with ARDS, each increase in LUS score was associated with an increase in lung density measured by chest CT. However, LUS score variations were not associated with lung recruitment [6]. Two other studies, one in patients with pneumonia and one in patients with ARDS [5,12], showed that both the LUS score and the reaeration score correlated well with changes in lung aeration quantified by chest CT. This was confirmed in a recent study in patients with ARDS caused by coronavirus disease 2019 [16]. In ventilated surgical patients without lung injury, LUS has been used successfully for semi–quantification of changes in lung aeration [17,18]. The results of a larger study in orthopedic surgery patients are eagerly awaited [19]. Finally, in one randomized clinal trial in patients receiving intraoperative ventilation, LUS was capable of detecting transient changes in aeration [20].

It should be noted that patients in the higher PEEP group had a higher cumulative fluid balance at the moment of the second LUS examination. This may have caused an increase in hydrostatic edema, which caused higher LUS scores [21,22,23]. This may have blurred our findings. Not controlling the fluid balances could be seen as a limitation of our study.

The study has other limitations. We were unable to perform LUS before randomization and start of ventilation with lower or higher PEEP. It should be noted that ventilation according to randomization had to start within one hour after start of ventilation in the ICU. This time window was too short to obtain informed consent for this substudy, and thus to perform earlier LUS. In addition, we performed LUS only in the first 24 to 48 h after start of ventilation, hence we cannot exclude changes, and difference in changes in LUS between the two groups after this timepoint. Second, we did not control other ventilator settings that may also affect lung aeration. Third, this was a single center study, and due to the block randomization, there was ultimately an imbalance between the number of patients in the two groups. Fourth, we did not use the gold standard (e.g., CT scan) method to quantify pulmonary aeration, in order to definitively assess the exact changes in pulmonary air volumes. In addition, scores were assessed by only one sonographer at a time, without calculation of interobserver or intraobserver variability. Finally, we could not perform a baseline ultrasound before randomization.

In our secondary analysis of a randomized clinical trial comparing a higher PEEP with a lower PEEP strategy in invasively ventilated ICU patients without ARDS, although the LUS scores increased over time in lower PEEP group and decreased at the same time in the higher PEEP group, differences between the groups were non–significant. This trend may assume significance between groups performing LUS in a wider cohort or at different timepoints.

## 5. Conclusions

In this substudy of a randomized clinical trial comparing lower versus higher PEEP in invasively ventilated patients without ARDS, LUS was not capable of detecting significant differences in changes in aeration. Future studies may validate our results in different clinical settings and in a broader population of patients to enhance generalizability and to further investigate the role of LUS in the assessment of the extent of pulmonary loss of aeration.

## Figures and Tables

**Figure 1 diagnostics-13-01989-f001:**
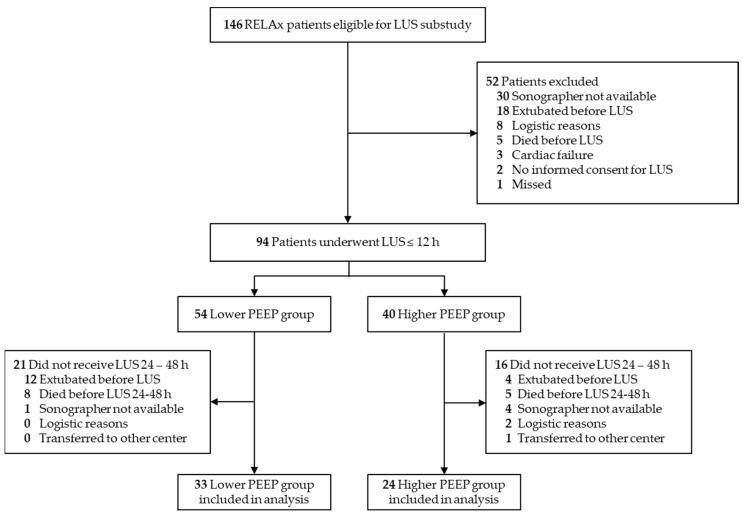
Patients flowchart.

**Figure 2 diagnostics-13-01989-f002:**
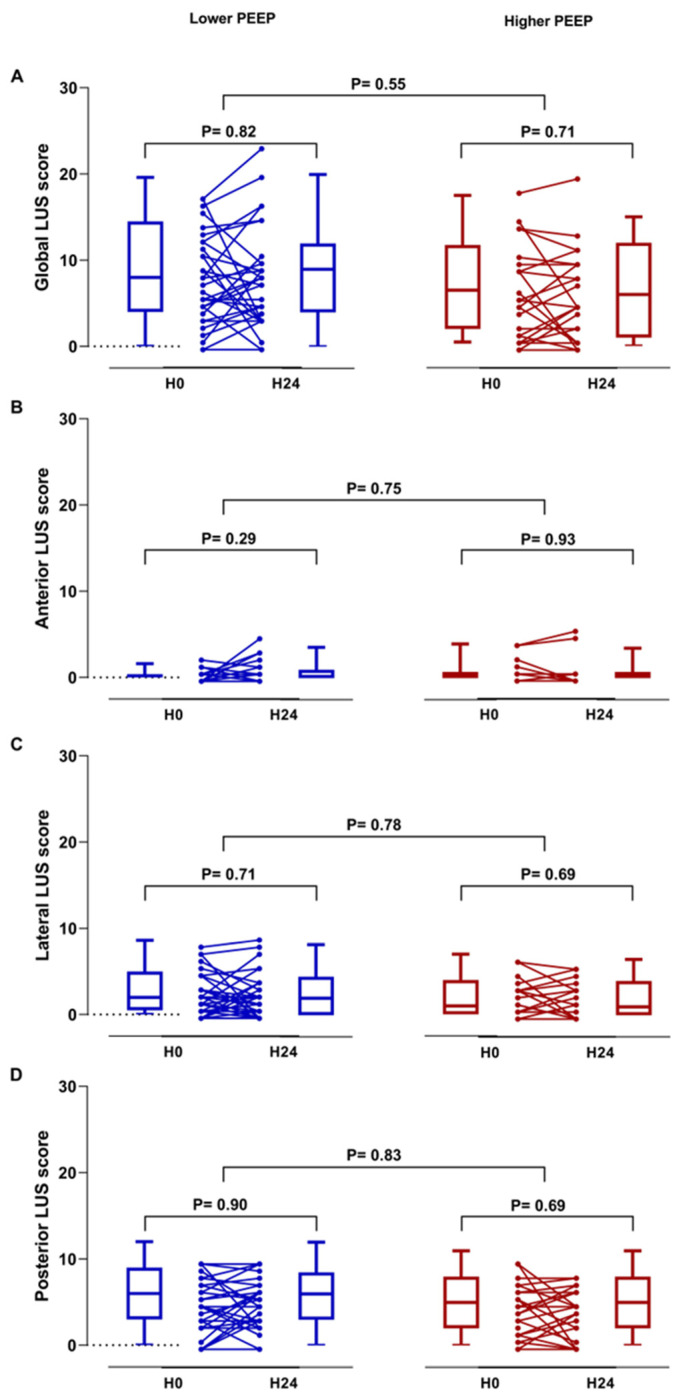
The change in global (**A**) and regional LUS scores (**B**–**D**) in patients ventilated with lower and higher PEEP strategies.

**Table 1 diagnostics-13-01989-t001:** Baseline Characteristics.

	Lower PEEP (*n* = 33)	Higher PEEP (*n* = 24)	*p* Value
**Demographics**			
Age	64 (52–72)	66 (59–71)	0.94
Female	14 (42%)	6 (25%)	0.28
Weight, kg	80 (71–89)	81 (68–91)	0.94
Height, cm	174 (168–184)	178 (172–180)	0.45
BMI, kg/height in m^2^	28 (23–31)	25 (66–75)	0.38
Fluid balance, in ml	387 (−106–2007)	945 (−45–3011)	0.35
APACHE II ^a^	24 (17–27)	25 (20–28)	0.37
APACHE IV ^b^	77 (56–92)	89 (63–102)	0.37
SOFA ^c^	10 (7–12)	10 (8–12)	0.42
**Comorbidities**			
Myocardial infarct	4 (12%)	4 (17%)	0.65
Heart failure	2 (6%)	0 (0%)	0.62
AKI	6 (18%)	1 (4.2%)	0.24
Surgical procedure in last 7 days	13 (39%)	7 (29%)	0.60
Stroke	10 (30%)	4 (17%)	0.38
Neurologic pulmonary edema	0 (0%)	0 (0%)	NA
Pancreatitis	0 (0%)	1 (4%)	0.87
None	6 (18%)	11 (46%)	0.82
**Reason for intubation**			0.82
Respiratory failure	10 (30%)	9 (38%)	
Pneumonia	4 (12%)	5 (21%)	
Cardiogenic pulmonary edema	3 (9%)	1 (5%)	
Sepsis non pulmonary	2 (6%)	1 (5%)	
COPD	0 (0%)	1 (5%)	
Other cause respiratory failure	1(3%)	1 (5%)	
Depressed consciousness	10 (30%)	7 (29%)	
OHCA	4 (12%)	4 (17%)	
Planned ventilation post-surgery	6 (18%)	2 (8%)	
Securing airway	2 (6%)	2 (8%)	
Trauma	1 (3%)	0 (0%)	

Data are given as median with IQR. Numbers are presented with (%). Abbreviations: AKI, acute kidney injury; APACHE, Acute Physiology and Chronic Health Evaluation; BMI, body mass index calculated as weight in kilograms divided by height in meters squared; COPD, chronic obstructive pulmonary disease; IQR, interquartile range; SOFA, Sequential Organ Failure Assessment. ^a^ APACHE IV score ranges from 0 to 286, with higher scores indicating more severe disease and a higher risk of death. ^b^ APACHE II score ranges from 0 to 71, with higher scores indicating more severe disease and a higher risk of death. ^c^ SOFA score ranges from 0 to 24, with higher values indicating a more severe condition.

**Table 2 diagnostics-13-01989-t002:** Respiratory and hemodynamic parameters during both LUS examinations.

	LUS 12 h			LUS 24–48 h		
	Lower PEEP (*n* = 33)	Higher PEEP (*n* = 24)	*p* Value	Lower PEEP (*n* = 33)	Higher PEEP (*n* = 24)	*p* Value
**Mode of Ventilation**			0.54			0.06
PCV	14 (42%)	14 (58%)		6 (18%)	9 (38%)	
VCV	2 (6%)	1 (4%)		0 (0%)	0 (0%)	
PSV	14 (42%)	6 (25%)		26 (79%)	12 (50%)	
ASV	3 (9%)	3 (13%)		1 (3%)	3 (13%)	
**Ventilatory variables**						
Pmax, cm H_2_O	15 (13–21)	22 (19–24)	<0.001	13 (10–17)	19 (16–21)	<0.001
PEEP, cm H_2_O	1.0 (0.0–5.0)	8.0 (8.0–8.0)	<0.001	0 (0–1)	8 (8–8)	<0.001
FiO_2,_ %	40 (30–50)	30 (24–40)	0.03	30 (25–40)	25 (21–35)	0.05
SpO_2,_ %	98 (97–100)	98 (96–99)	0.34	96 (94–98)	96 (94–99)	0.91
PaO_2_/FiO_2_ ratio	281 (210–375)	314 (226–403)	0.28	255 (201–318)	334 (208–393)	0.17
Tidal volume, mL/kg PBW	7.0 (5.8–8.1)	6.7 (6.1–7.3)	0.65	7.5 (6.6–8.3)	6.4 (5.6–8.9)	0.31
Respiratory rate, breaths/min	18 (17–22)	20 (18–22)	0.56	18 (15–22)	20 (17–25)	0.12
Minute volume, L/min	8.7 (8–10)	9.9 (8–11)	0.42	9 (8–11)	10 (9–12)	0.14
**Blood gas**						
pH	7.4 (7.4–7.5)	7.4 (7.4–7.5)	0.88	7.4 (7.4–7.5)	7.4 (7.4–7.5)	0.44
PaO_2_, mmHg	98 (84–113)	89 (82–113)	0.69	81 (67–97)	83 (77–88)	0.90
PaCO_2_, mmHg	38 (33–42)	37 (32–42)	0.67	39 (35–42)	36 (32–41)	0.42
**Hemodynamics**						
Heart rate, bpm	84 (67–96)	83 (70–94)	0.79	81 (67–97)	82 (66–98)	0.88
MAP, mmHg	78 (71–91)	77 (72–83)	0.99	78 (72–86)	87 (75–91)	0.34
SBP, mmHg	112 (102–137)	112 (101–120)	0.50	122 (109–147)	125 (104–140)	0.73
DBP, mmHg	59 (53–67)	60 (56–70)	0.29	59 (51–67)	65 (56–69)	0.20
Fluid balance, mL	387 (−106–2007)	945 (−45–3011)	0.35	803 (−124–1666)	1610 (874–2528)	0.02

Data are given as median with IQR. Numbers are presented with (%). Abbreviations: ASV, adaptive support ventilation; bpm, beats per minute; DBP, diastolic blood pressure; FiO_2_, fraction of inspired oxygen; IQR, interquartile range; MAP, mean arterial blood pressure; PaCO_2_, partial pressure of carbon dioxide; PaO_2_, partial pressure of arterial oxygen; PCV, pressure controlled ventilation; PEEP, positive end-expiratory pressure; Pmax, maximum airway pressure; PSV, pressure support ventilation; SBP, systolic blood pressure; SpO_2_ oxygen saturation as measured by pulse oximetry; VCV, volume controlled ventilation.

**Table 3 diagnostics-13-01989-t003:** Lung ultrasound signs of atelectasis in the two PEEP groups. Numerical scores were calculated for subpleural consolidations, static air bronchograms and B-lines—each lung ultrasound region with the presence of SPC, SAB or B-pattern was assigned a 1 and all 12 zones were summed resulting in a score from 0 to 12 per patient, with higher scores indicating more severe atelectasis.

	LUS 12 h			LUS 24–48 h		
	Lower PEEP (*n* = 33)	Higher PEEP (*n* = 24)	*p* Value	Lower PEEP (*n* = 33)	Higher PEEP (*n* = 24)	*p* Value
**Subpleural consolidation**						
Global						
All regions	1 (0–3)	2 (0–2)	0.80	1 (0–2)	1 (0–2)	0.45
Regional						
Anterior	0 (0–0)	0 (0–0)	0.88	0 (0–0)	0 (0–0)	1.00
Lateral	0 (0–1)	0 (0–1)	0.97	0 (0–1)	0 (0–1)	0.50
Posterior	1 (0–2)	1 (0–1)	0.45	1 (0–1)	0 (0–1)	0.27
**B-lines**						
Global						
All regions	3 (0–5)	3 (1–4)	0.85	3 (1–6)	2 (1–5)	0.56
Regional						
Anterior	0 (0–0)	0 (0–0)	0.94	0 (0–1)	0 (0–0)	0.60
Lateral	1 (0–2)	1 (0–2)	0.84	1 (0–2)	1 (0–2)	0.29
Posterior	2 (0–3)	2 (1–3)	0.89	2 (0–3)	2 (0–2)	0.90
**Static Air Bronchogram**						
Global						
All regions	0 (0–0)	0 (0–0)	0.96	0 (0–0)	0 (0–0)	0.74
Regional						
Anterior	0 (0–0)	0 (0–0)	NA	0 (0–0)	0 (0–0)	0.39
Lateral	0 (0–0)	0 (0–0)	0.47	0 (0–0)	0 (0–0)	NA
Posterior	0 (0–0)	0 (0–0)	0.69	0 (0–0)	0 (0–0)	0.54

Data are given as median with IQR. Abbreviations: SAB, static air bronchogram; SPC, subpleural consolidation; PEEP, positive end-expiratory pressure.

## Data Availability

Requests for the data should be sent to Claudio Zimatore; email address: claudiozimatore@gmail.com.

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
