# Peer review of "Lung Ultrasound to Determine the Effect of Lower vs. Higher PEEP on Lung Aeration in Patients without ARDS—A Substudy of a Randomized Clinical Trial"

_diagnostics, 2023, doi:10.3390/diagnostics13121989_

Round 1
Reviewer 1 Report
Thank you for the opportunity to review your manuscript. The idea of using bedside LUS to help assess the impact of PEEP on lung recruitment/aeration in non-ARDS patients is very interesting.
Why did the authors choose 8mmHg as the cutoff between the high and low PEEP group?
Did the investigators include patients that were in the prone ?
It is understandable that obtaining a CT scan within 12hours and then at 48hrs of invasive ventilation is difficult to obtain , without this gold standard establishing the success of recruitment between the two groups would be hard. Were there other data points that could have been to highlight this phenomenon for example comparing the FiO2 among the low PEEP at 12 hrs and 48 hrs (and doing the same for the high PEEP group) may offer some insight.
Do we know why did the patients in the high PEEP group receive more fluids? This certainly would have impacted your results
Would like to the p values or CI for each Ventilation mode . PSV use in the Lower PEEP group is much higher than the high PEEP group
Based on table 3 is it fair to assume that no static air bronchograms were visualized in any of the subjects?
How did the authors deal with pathologies like PNA and cardiogenic pulm edema that can create LUS findings similar to the ones used calculating the LUS score. Did authors look for dynamic air bronchograms ? What about the lobes that were infected with PNA ? how were we sure that the static air bronchograms weren't sec to the infectious process and not atelectasis?
Can the authors elaborate more on how randomization of the patients took place as well as how blinding of the ultrasound operator took place (was the ventilator machine covered before they entered the room ?)
The statistical model used doesn't seem to take into account several covariates that can affect the results for example: accounting for the difference in fluid balance between the groups, accounting for the underlying lung pathology or method of ventilation. I would urge the authors to consider a logistic regression model or even better a mixed effects multilevel model to account for these variables as well as the possible clustering effect inherent to repeated measures on the same patient .

The English quality is very good very few and minor mistakes
Reviewer 2 Report
The current study titled “Lung Ultrasound to Determine the Effect of Lower vs Higher PEEP on Lung Aeration in Patients without ARDS––a substudy of a randomized clinical trial” Ref: diagnostics-2397219, deals with an important subject. Fair results and observations were mentioned concerning the studied cases. Minor revisions are needed.
- Due to what mentioned in the last paragraph of page 9 as rational for the attained results, the author(s) should add a short paragraph for what should be made to modify the methodology and get better results.
- The conclusion section should be modified explaining the entire observation/results attained and also give short notes/suggestions for future studies.
Reviewer 3 Report
- why the pressure if eight was chosen for high PEEP strategy
- during the article are used both aeration and rearation scores. Do these terms reflect the same notion?
minor revisions
Round 2
Reviewer 1 Report
I have no further comments to make, thank you authors for your detailed response
English quality was excellent